# The Blockade of Mitogen-Activated Protein Kinase 14 Activation by Marine Natural Product Crassolide Triggers ICD in Tumor Cells and Stimulates Anti-Tumor Immunity

**DOI:** 10.3390/md21040225

**Published:** 2023-03-31

**Authors:** Keng-Chang Tsai, Chia-Sheng Chen, Jui-Hsin Su, Yu-Ching Lee, Yu-Hwei Tseng, Wen-Chi Wei

**Affiliations:** 1National Research Institute of Chinese Medicine, Ministry of Health and Welfare, Taipei 112026, Taiwan; 2Ph.D. Program in Medical Biotechnology, College of Medical Science and Technology, Taipei Medical University, Taipei 110301, Taiwan; 3Department of Psychiatry, Kaohsiung Armed Forces General Hospital, Kaohsiung 80284, Taiwan; 4National Museum of Marine Biology and Aquarium, Pingtung 94450, Taiwan; 5Department of Marine Biotechnology and Resources, National Sun Yat-sen University, Kaohsiung 804201, Taiwan; 6TMU Research Center of Cancer Translational Medicine, Taipei Medical University, Taipei 110301, Taiwan

**Keywords:** immunogenic cell death, damage-associated molecular patterns, mitogen-activated protein kinase 14, natural product, crassolide

## Abstract

Immunogenic cell death (ICD) refers to a type of cell death that stimulates immune responses. It is characterized by the surface exposure of damage-associated molecular patterns (DAMPs), which can facilitate the uptake of antigens by dendritic cells (DCs) and stimulate DC activation, resulting in T cell immunity. The activation of immune responses through ICD has been proposed as a promising approach for cancer immunotherapy. The marine natural product crassolide, a cembranolide isolated from the Formosan soft coral *Lobophytum michaelae*, has been shown to have cytotoxic effects on cancer cells. In this study, we investigated the effects of crassolide on the induction of ICD, the expression of immune checkpoint molecules and cell adhesion molecules, as well as tumor growth in a murine 4T1 mammary carcinoma model. Immunofluorescence staining for DAMP ectolocalization, Western blotting for protein expression and Z′-LYTE kinase assay for kinase activity were performed. The results showed that crassolide significantly increased ICD and slightly decreased the expression level of CD24 on the surface of murine mammary carcinoma cells. An orthotopic tumor engraftment of 4T1 carcinoma cells indicated that crassolide-treated tumor cell lysates stimulate anti-tumor immunity against tumor growth. Crassolide was also found to be a blocker of mitogen-activated protein kinase 14 activation. This study highlights the immunotherapeutic effects of crassolide on the activation of anticancer immune responses and suggests the potential clinical use of crassolide as a novel treatment for breast cancer.

## 1. Introduction

The World Health Organization recently estimated that in 2020, 2.3 million women were diagnosed with breast cancer and there were approximately 685,000 deaths worldwide. By the end of that year, 7.8 million women who had been diagnosed with breast cancer within the previous five years were still alive, making it the most common cancer globally [1,2].

Unlike conventional chemotherapy, which destroys malignant cells, cancer immunotherapy is a treatment that boosts the body’s immune system to fight cancer. Immunotherapies that have shown sustained anticancer responses in several cancers hold promise for researchers and patients. Currently, identifying and developing potential targets to enhance anticancer responses is a key strategy in cancer immunotherapy. For example, immune checkpoint molecules like PD-1 and CTLA-4 have been identified as negative regulators of immune activation [3,4,5]. The binding of PD-L1 on cancer cells to PD-1 on T cells suppresses the activation and function of T cells [6,7]. In addition to immune checkpoints, cancer cells can also express other “don’t eat me” signals, such as CD47 and CD24, on their surface to protect themselves from immune surveillance [8,9]. Despite the promising results of immunologic checkpoint blockade with antibodies, the objective response rate of clinical drugs still needs further improvement.

On the other hand, immunogenic cell death (ICD) refers to a type of cell death that stimulates immune responses. ICD is characterized by the surface exposure of damage-associated molecular patterns (DAMPs), a group of endogenous danger molecules that includes heat shock proteins (HSPs), glucose-related protein (GRP), HMGB1, calreticulin (CRT), and others [10,11]. These molecules can facilitate the uptake of antigens by dendritic cells (DCs) and stimulate DC activation, resulting in T cell immunity. Activation of immune responses through ICD has been proposed as a promising approach for cancer immunotherapy. Currently, a number of ICD-induced chemotherapeutic agents are being used clinically for cancer treatment [11]. 

Natural products from marine organisms are considered key sources for drug development [12]. For example, marine sponge natural products with anti-cancer activity have been developed into drugs [13]. However, research on the immunotherapeutic activity of marine natural products for cancer treatment is still limited. The marine natural product crassolide, a cembranolide isolated from the Formosan soft coral *Lobophytum michaelae*, has been shown to have cytotoxic effects on cancer cells [14]. Cytotoxic activities related to G2/M phase arrest and apoptosis were also reported [15]. In this study, we investigated the effects of crassolide on the induction of ICD, the expression of immune checkpoint molecules and cell adhesion molecules, as well as tumor growth in a murine 4T1 mammary carcinoma model. Additionally, we identified its target molecules. This study highlights the immunotherapeutic effects of crassolide on the activation of anti-cancer immune responses and suggests the potential clinical use of crassolide as a novel treatment for breast and other types of cancer.

## 2. Results

### 2.1. Crassolide Stimulated the Cytotoxicity against Human Breast Cancer Cells and Murine Mammary Carcinoma Cells

The structure of crassolide is shown in Figure 1. To determine the cytotoxic effects of crassolide on different types of mammary tumor cells, we used a methylthiazole tetrazolium (MTT) assay to measure the cell viability of test cells treated with crassolide at different concentrations for 24 h. The results showed that crassolide inhibits the cell viability of the four mammary carcinoma cells in a dose-dependent manner, as shown in Figure 2. The IC50 values of cell viability in MCF7, MDA-MB-231, 4T1-luc2, and TS/A cells were 9.35, 6.69, 24.6, and 3.74 µM, respectively (Table 1).

### 2.2. Crassolide Significantly Increases the Expression of DAMPs on the Surface of 4T1-luc2 Murine Mammary Carcinoma Cells

Immunogenic cell death (ICD) of tumor cells, characterized by surface expression of damage-associated molecular patterns (DAMPs), stimulates antitumor immunity. HSP70, HSP90, HMGB1, and calreticulin (CRT) have been recognized as key DAMP molecules. Crassolide exhibited cytotoxicity in murine mammary carcinoma cells 4T1-luc2. We then investigated if crassolide can stimulate ICD in 4T1-luc2 cells by measuring the expression levels of HSP 70, HSP 90, HMGB1, and CRT on the surface of 4T1-luc2 cells that were incubated in different concentrations (50 to 3.125 µM) of crassolide for 24 h. The surface expression of HSP70, HSP90, HMGB1, and CRT were then measured by flow cytometry analysis. Figure 3A showed that the 4T1-luc2 cells’ control group expressed 17.51% of HSP90, 9.16% of HSP70, 5.99% of HMGB1, and 19.56% of CRT. The 4T1-luc2 cells treated with crassolide had a substantial expression level of HSP90, reaching the highest peak at 25 µM treatment (50.08%). The highest expression level of HSP70 (29.10%) appeared with 12.5 µM treatment. Crassolide increased the expression of HMGB1 and CRT in a dose-dependent manner, with HMGB1 (96.48%) and CRT (78.68%) reaching the highest expression at 50 µM treatment, respectively. Crassolide-treated tumor cells were visualized for expression of ectoDAMPs by confocal microscopy through indirect immunofluorescent staining. The expression of HSP70, HSP90, HMGB1 and CRT was significantly increased in crassolide-treated tumor cells compared to untreated tumor cells (Figure 3B). These results indicate that crassolide significantly increases immunogenic cell death (ICD) in murine mammary carcinoma cells 4T1-luc2.

### 2.3. Crassolide Enhances Expression of Immune Checkpoint Molecule PD-L1 but Inhibits the Expression of Heat-Stable Antigen/CD24 on the Surface of 4T1-luc2

Our data showed that crassolide significantly induced immunogenic cell death (ICD) in 4T1-luc2 cells. We further examined the effect of crassolide on the regulation of “don’t eat me signals” in these cells. The cells were treated with various concentrations of crassolide for 24 h, and we used flow cytometry analysis to measure the expression of PD-L1 and CD24 on the cell surface. Figure 4A illustrates that the control group of 4T1-luc2 cells expressed 48.94% PD-L1 and 74.56% CD24. Crassolide slightly decreased CD24 expression, but it significantly increased PD-L1 expression, with the highest peak at 50 µM treatment (86.71%). Additionally, the immunofluorescence image confirmed that crassolide enhances PD-L1 expression and suppresses CD24 expression on the surface of 4T1-luc2 cells (Figure 4B).

### 2.4. Crassolide Significantly Induces the Translocation of DAMPs to the Surface of 4T1-luc2 Cells

Crassolide significantly increased the expression levels of DAMPs and PD-L1 on the surface of 4T1-luc2 cells. To understand the molecular mechanisms underlying the regulatory effect of crassolide on the expression of DAMPs, CD24, and PD-L1, we evaluated whether crassolide can stimulate their expression in 4T1-luc2 cells. The 4T1-luc2 cells were incubated with different concentrations of crassolide for 24 h, and the total protein was collected for assessing the expression of HSP70, HSP90, CRT, and HMGB1 using Western blot assay. Figure 5A shows that crassolide suppressed the expression of HSP70, HSP90, and CRT, but had no effect on the expression of HMGB1 protein. Similarly, Figure 5B shows that crassolide suppressed the expression of CD24 and PD-L1. Taking into account the results from Figure 3, Figure 4 and Figure 5, it can be inferred that crassolide suppressed the protein expression of DAMPs and PD-L1 but promoted the translocation of DAMPs and PD-L1 to the surface of 4T1-luc2 cells.

### 2.5. Crassolide-Treated 4T1-luc2 Cells Effectively Immunized Mice against Primary Tumors

To assess the ability of the ICD inducer to trigger antitumor immune responses, one method is to inject mice with in vitro killed tumor cells treated with the ICD inducer, followed by a challenge with untreated tumor cells [16]. To determine if crassolide-treated 4T1-luc2 cells can elicit antitumor immunity in mice with tumors, the mice were subcutaneously given 4T1-luc2 cells that underwent freeze-thaw cycles or treated with crassolide at 25 µM. The 4T1-luc2 tumor cells were implanted into the mammary fat pad orthotopically after 7 days of post-vaccination. The treatment of 4T1-luc2 cells with Crassolide seemed to suppress tumor growth, suggesting that it can induce anti-tumor immunity. (Figure 6A). Bioluminescence imaging (BLI) and quantitative data of BLI confirmed the effect, showing decreased activity in tumor cells. The p-value was less than 0.05 (Figure 6B,C). The body weight loss of treated mice was not impacted by the vaccine (Figure 6D).

### 2.6. Crassolide Inhibited MAPK14 Kinase Activity

The target protein of crassolide was identified through a Discovery Studio pharmacophore reverse docking. The results showed MAPK14 kinase as a potential target protein. MAPK14, known as p38α kinase, is made up of two distinct parts, the smaller N-terminal lobe which is primarily composed of β-sheets, and the α-helical C-terminal lobe. The two lobes are connected by a flexible hinge that forms the ATP-binding site, and structural elements from both lobes contribute to this site. During the canonical activation pathway, MAP2Ks phosphorylate the TGY sequence in the activation loop, leading to significant structural changes and the creation of a distinctive β-sheet motif that is located away from both the ATP- and substrate-binding sites [17]. Figure 7A,B depicted crassolide docked with MAPK14 kinase. Crassolide established interactions with Thr106 and Lys53 through hydrogen bonds and with Leu167, Met109, Ile84, Ala51, and Val38 residues in the active site through hydrophobic interactions. The inhibitory effect of crassolide on MAPK14 kinase activity was confirmed in a cell-free system using a biochemical method (Z′-LYTE Kinase Assay). Results showed crassolide could concentration-dependently suppress MAPK14 kinase activity with an IC50 value of approximately 5.15 µM (Figure 7C).

### 2.7. Crassolide Up-Regulated Phosphorylation of MAPK14 and Its Downstream Targets in 4T1-luc2 Cells

The regulation of MAPK14 kinase activity is controlled by phosphorylation at two sites (Thr180/Tyr182) [17]. The effect of crassolide on the phosphorylation status of MAPK14 at Thr180 and Tyr182 was assessed in 4T1-luc2 cells. Results showed that crassolide upregulated the phosphorylation of Thr180 and Tyr182 in a dose-dependent manner (Figure 8). The effect of crassolide on the phosphorylation of downstream proteins of MAPK14 such as NF-κB, STAT1, and EIK-1 was also measured. Results showed that crassolide downregulated the phosphorylation of NF-κB, STAT1, and EIK-1 in 4T1-luc2 cells (Figure 8).

## 3. Discussion

Immunogenic cell death (ICD) and damage-associated molecular patterns (DAMPs) are emerging as promising strategies in cancer therapy. ICD triggers an immune response against cancer cells, while DAMPs are molecules released by dying or damaged cells that activate the immune system. These mechanisms are being utilized to improve the efficacy of cancer immunotherapy. ICD and DAMPs have shown promising results in preclinical and clinical studies [11,18] and are being evaluated in combination with other immunotherapeutic approaches such as checkpoint inhibitors and vaccines [19,20,21]. ICD is triggered by different stressors, including endoplasmic reticulum (ER) stress and reactive oxygen species (ROS) production, and leads to the exposure of damage-associated molecular patterns (DAMPs) on the cell surface [22,23,24]. During the ICD process, immunogenic dead cells expose hallmarks on the cell surface and release substances that interact with immune cells and mediate immunogenicity. These hallmarks include calreticulin (CRT), ATP, and high mobility group B1 (HMGB1). ICD inducers, such as anthracyclines, PP1/GADD34 inhibitors, cardiac glycosides, oxaliplatin, bleomycin, cyclophosphamide, and shikonin, have been used to induce ICD and inhibit tumor growth [10,11]. Currently, only a few ICD inducers are being used in the clinic, but the development of new ICD inducers is expected to improve cancer treatment outcomes in the future.

The mitogen-activated protein kinases (MAPKs) are a highly conserved family of serine/threonine kinases that play a central role in a range of fundamental cellular processes such as cell growth, proliferation, death, and differentiation [25,26,27]. There are at least three distinct MAPK signaling modules in mammals that mediate extracellular signals into the nucleus to turn on responsive genes: ERK, JNK, and p38 kinase [26]. The MAPK pathway is essential in regulating many cellular processes, including inflammation, cell stress response, cell differentiation, and gene expression [27]. In the context of immunogenic cell death (ICD), the p38 MAPK pathway has been shown to be involved in the regulation of ER stress, which is a major causative agent of ICD [28]. In some cases, activation of p38 MAPK has been shown to promote ICD by inducing the release of danger signals and activating the immune system. The precise role of p38 MAPK in ICD is still not fully understood. There are four identified p38 MAP kinases: p38-α (MAPK14), p38-β (MAPK11), p38-γ (MAPK12/ERK6), and p38-δ (MAPK13/SAPK4) [27]. In this study, crassolide from marine soft coral was identified as a novel inhibitor of MAPK14. Crassolide not only significantly reduced the viability of human breast cancer cells and murine mammary carcinoma cells, but also increased ICD and decreased CD24 expression in the latter. Orthotopic tumor engraftment experiments showed that crassolide-treated tumors elicited anti-tumor immunity. These findings deepen our understanding of MAPK14’s role in ICD and highlight crassolide’s potential as a treatment for breast and other types of cancer by activating anti-cancer immune responses (Figure 9).

Crassolide was first found to have cytotoxicity in cancer cells [14]. Later, its ability to induce G2/M cell cycle arrest, apoptosis, and autophagy in cancer cells was investigated [15]. Based on observation of a crassolide-induced increase in phosphorylation of ERK, it was suggested that this capability was induced through ROS-mediated ER stress pathways [15]. Additionally, crassolide was also found to suppress dendritic cell maturation and attenuate experimental antiphospholipid syndrome. It was also found that, when applied at doses of only 10 and 20 mg/kg, crassolide produced hematotoxicity, hepatotoxicity, and renotoxicity, but these effects were not observed at a dose of 5 mg/kg [29]. In this study, we showed that crassolide inhibited p38α activity using a biochemical method (Figure 7C). Interestingly, crassolide upregulated the phosphorylation of p38α, known as its initial activation marker. In contrast, crassolide downregulated the phosphorylation of NF-κB, STAT1, and EIK-1, known as downstream effectors of p38α signaling transduction (Figure 8). Based on the research results and the similarity of their effects, we believe that crassolide is a novel p38 catalytic inhibitor (e.g., SB203580). Additionally, due to its structural similarity with ERK, we hypothesize that crassolide may also block ERK activity, resulting in an accumulation of ERK phosphorylation. However, further experiments are needed to confirm this hypothesis.

The abundance of diverse marine life and their distinct biochemical profiles have led to the discovery of a multitude of biologically active natural products derived from marine organisms, making them a valuable source for the development of new drugs. Despite this potential, the process of discovering and utilizing these products can be challenging due to difficulties in cultivating marine organisms and the complex structure of their active compounds, leading to limited yields. Our study highlights a novel approach by showcasing the effectiveness of the marine compound crassolide in stimulating immunogenic cell death in tumor cells in vitro and eliciting anti-tumor immunity in vivo when used as a vaccine, demonstrating the potential of marine natural products for clinical applications. However, just as natural product research in the ocean may face limitations in its applications, the modification of crassolide to enhance its binding affinity with MPAK14 and the synthesis of crassolide through total or semi-synthesis may also face limitations due to the complexity of the compound’s structure, which may restrict future applications and development. 

## 4. Materials and Methods

### 4.1. Origin of Crassolide

Crassolide was obtained from Dr. Jui-Hsin Su at the National Museum of Marine Biology and Aquarium in Pingtung, Taiwan, and was isolated from the wild-type soft coral Lobophytum crissum. 

### 4.2. Cell Lines

MCF7, MDA-MB-231 and TS/A cells were procured from the Food Industry Research and Development Institute (Hsinchu, Taiwan). The 4T1-luc2 cells, obtained from Dr. Pei-Wen Hsiao at Academia Sinica in Taipei, Taiwan, were developed from mouse mammary tumor 4T1 cells (ATCC, CRL-2539) and stably transfected with a luciferase transgene. These cells were cultured in RPMI-1640 (Invitrogen, 31800) supplemented with 10% fetal bovine serum (FBS) (GibcoTM, 10082147), 1 mM penicillin-streptomycin (GibcoTM, 15140122), MEM-Non-Essential Amino Acids Solution (NEAA) (ThermoFisher, 11140050), and 1 mM sodium pyruvate (GibcoTM, 11360070) at 37 °C in 5% CO_2_ and 95% humidity.

### 4.3. Mice

Female 8-week-old BALB/c mice were purchased from the National Laboratory Animal Breeding and Research Center in Taipei, Taiwan. All mice were treated with standard environmental and food conditions, including a temperature of 22 ± 1 °C, 55 ± 5% humidity, and a 12-h light/dark cycle, and had unlimited access to food and water at the National Research Institute of Chinese Medicine (MOHW). The mice were randomly assigned to four groups to reduce experimental bias. The experimental procedures involving animals were conducted in accordance with the Guide for the Care and Use of Laboratory Animals (National Research Council, 2011) and were approved by the Animal Research Committee of the National Research Institute of Chinese Medicine (ref no: NRICM-IACUC-109-355-1).

### 4.4. Animal Model

For prophylactic vaccination, the mice (*n* = 15) were randomly assigned to 3 groups: Group 1 (*n* = 5) served as the control group and was injected with PBS. Group 2 (*n* = 5) was injected with 2 × 10^6^ of 4T1-luc2 cells that had been put through freeze and thaw (F/T) cycles, where the cells were first frozen in liquid nitrogen for 90 s, then thawed at 4 °C for 4 min, and this process was repeated 4 times. Group 3 (*n* = 5) was injected with 2 × 10^6^ of 4T1-luc2 cells that had been treated with crassolide (25 µM) for 24 h. Seven days post-vaccination, all four groups of mice were orthotopically implanted with 5 × 10^5^ live 4T1-luc2 tumor cells in 100 µL of PBS into the mammary fat pad. The tumor size was observed and measured three times a week and bioluminescence imaging of tumors was monitored once a week. Tumor size was measured by caliper measurements, and tumor volume calculation was used following formula: 0.5 × length (mm) × width^2^ (mm^2^). Bioluminescence imaging of tumors was monitored using a noninvasive in vivo imaging system. The mice were injected intraperitoneally with 15 mg/kg of D-luciferin potassium (PerkinElmer, 122799) in PBS and were left for 5 min before imaging. The test mice were then anesthetized with 2.5% isoflurane using the XGI-8 Gas Anesthesia System (PerkinElmer, Boston, MA, USA), and the XENOGEN IVIS 50 (PerkinElmer, Boston, MA, USA) was applied.

### 4.5. Cell Viability Assay

MCF7, MDA-MB-231, and 4T1-luc2 tumor cells (1 × 10^4^) were prepared and dispensed in vehicle or test compounds in 96-well plates. They were then washed with PBS and incubated in RPMI 1680, placed in a 5% CO_2_ incubator for 24 h. After this, the cells were treated with crassolide (50 to 3.125 µM) for 24 h. All treatments were performed in triplicate cell cultures. Cell viability was assayed using the 3-(4, 5-dimethylthiazol-2-yl)-2,5-diphenyltetrazolium bromide (MTT) colorimetric method. The optical density at absorbance 570 nm (A570) was measured using a multi-wall scanning spectrophotometer.

### 4.6. Immunofluorescence

4T1-luc2 tumor cells were cultured in 24-well plates with 0.1% poly-L-lysine coated coverslips. These cells were grown in RPMI 1640 medium. The cells were washed with PBS and then treated with crassolide (50 to 3.125 µM) for 24 h. For fluorescent immunostaining, the cells were fixed with cold methanol (−20 °C) and blocked with 3% bovine serum albumin (BSA, Bionovas) in PBS. Primary antibodies for heat shock protein 90 (HSP90, 1:400, Cell signaling), heat shock protein 70 (HSP70, 1:400, Abcam), high mobility group box 1 (HMGB1, 1:400, Cell signaling), calreticulin (CRT, 1:400, Cell signaling) were added to the cells, diluted in 3% BSA, 0.1% tween-20 in PBS, and protected from light. Finally, the cells’ nuclei were stained with 4,6-diamidino-2-phenylindole (DAPI, 5 μg/mL, Thermo Fisher Scientific, Inc.) for 5 min. To mount the expression of DAMPs on cells, the immunofluorescence staining protocol was applied, and then the cells were examined under a confocal microscope.

### 4.7. Detection of DAMPs Ectolocalization

The 4T1-luc2 tumor cells were treated with crassolide (50 to 3.125 µM) for 24 h. Aliquots of 1 × 10^6^ 4T1-luc2 tumor cells were harvested, washed with PBS, and stained with PE anti-HSP90 antibody (LSBio, LS-63265), Alexa Fluor 488 anti-Hsp70 antibody (BioLegend, #648004), PE anti-HMGB1 antibody (BioLegend, #651404), PE anti-calreticulin (Cell signaling, #62304), APC anti-PD-L1 (BioLegend, #124312), Alexa Fluor 488 anti-CD24 (BioLegend, #101816) in 0.5% bovine serum albumin (BSA, Bionovas) in PBS, and then analyzed using flow cytometry (BD FACSVerse, San Jose, CA, USA).

### 4.8. Western Blotting

To further investigate the effect of crassolide on tumor growth and the expression of immune checkpoint molecules, 4T1-luc2 tumor cells were treated with crassolide (at concentrations of 50 to 3.125 µM) for 1 h or 24 h. Aliquots of 1 × 10^6^ cells were harvested and washed with PBS, then lysed with RIPA lysis and extraction buffer. Protein content was determined using the Coomassie (Bradford) Protein Assay Kit. Samples were then fractionated by 10% SDS-PAGE and proteins were transferred onto a nitrocellulose membrane filter paper sandwich (ThermoFisher, LC2001). The 1 h group was immunoblotted with antibodies specific for phospho-p44/42 MAPK (Erk1/2) (Cell Signaling Technology, #9101), phospho-NF-κB p65 (Ser536) (Cell Signaling Technology, #3033), phospho-p38 (Cell Signaling Technology, #4511), phospho-JNK (Cell Signaling Technology, #4668), anti-ELK1 (Abcam, ab218133), phospho-STAT1 (Cell Signaling Technology, #9167), and β-actin (Cell Signaling Technology, #8457). The 24-h group was immunoblotted with antibodies specific for HSP90 (Cell Signaling Technology, #4877), anti-HSP70 (Abcam, ab181606), anti-calreticulin (Abcam, ab2907), anti-PD-L1 (Abcam, ab213480), anti-CD24 (Abcam, ab290745), HMGB1 (Cell Signaling Technology, #6893), and β-actin (Cell Signaling Technology, #8457). Non-specific binding was removed and target proteins were detected using corresponding secondary antibodies. Protein bands were then detected and developed using enhanced West Femto Maximum Sensitivity Substrate (ThermoFisher, 34096) and the Touch Imager (e-BLOT, Pudong, Shanghai, China).

### 4.9. Molecular Modeling of Crassolide and MAPK14 Interaction

To further investigate the binding interaction between crassolide and MAPK14 kinase, the crassolide was docked into the active site of MAPK14 using the GOLD docking tool in BIOVIA Discovery Studio 2021 (BIOVIA Corp., San Diego, CA, USA). The crystal structure of human MAPK14 (P38 kinase) was obtained from the RCSB Protein Data Bank (PDB ID: 1DI9) and the protein and crassolide compound atoms were assigned CHARMm force field.

### 4.10. Z′-LYTE Kinase Assay

An IC50 value of crassolide for inhibition of MAPK14 kinase activity was evaluated using a fluorescence resonance energy transfer-based Z′-LYTE kinase assay performed by Thermo Fisher Scientific’s SelectScreen™ Profiling Service. Briefly, the 2X MAPK14 (p38 alpha) direct/Ser/Thr 15 mixture was prepared in 50 mM HEPES (pH 6.5), 0.01% BRIJ-35, 10 mM MgCl2, 1 mM EGTA, and 0.02% NaN3. The final 10 µL kinase reaction consisted of 7.14–103 ng MAPK14 (p38 alpha) direct and 2 µM Ser/Thr 15 in 50 mM HEPES (pH 7.0), 0.01% BRIJ-35, 10 mM MgCl2, 1 mM EGTA, and 0.01% NaN3. After a 1-h kinase reaction incubation, the reaction was developed and terminated, and the fluorescence ratio was calculated according to the manufacturer’s protocol.

## 5. Conclusions

We have demonstrated that the marine natural product crassolide induces immunogenic cell death of tumor cells by blocking MPAK14 activation, thereby stimulating anti-tumor immunity. Our study not only highlights the immunotherapeutic effects of crassolide in activating antic ancer immune responses, but also provides a novel approach to developing marine natural products for clinical applications. 

## Figures and Tables

**Figure 1 marinedrugs-21-00225-f001:**
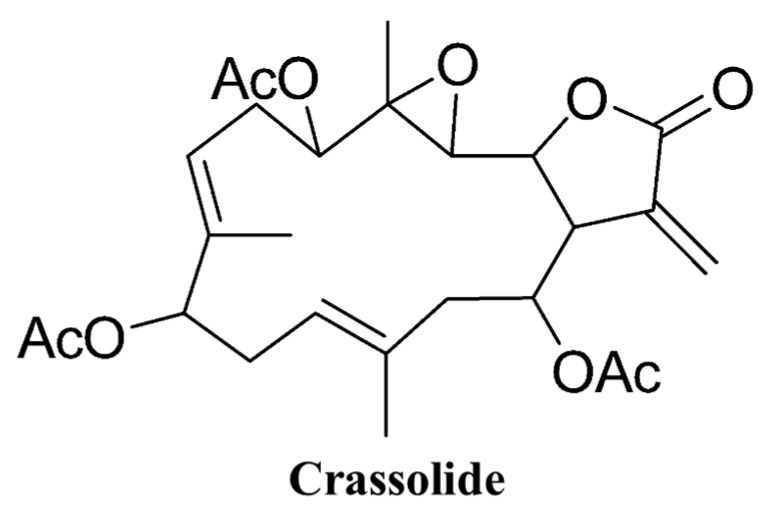
The chemical structure of crassolide.

**Figure 2 marinedrugs-21-00225-f002:**
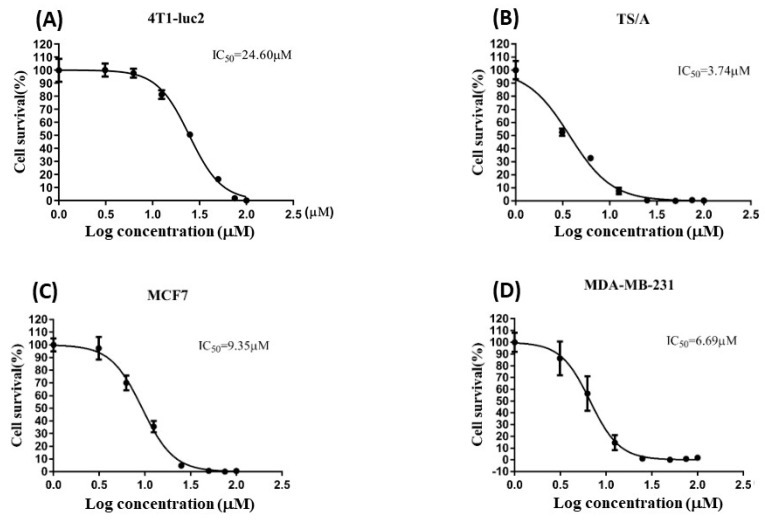
The cytotoxicity of crassolide in human breast cancer cells and murine mammary carcinoma cells. Cell lines 4T1-luc2 (**A**), TS/A (**B**), MCF7 (**C**) and MDA-MB-231 (**D**) were treated with increasing concentrations of crassolide for 24 h and cell survival was determined by MTT assay. Data are represented as the mean ± SD. The IC50 values of cell viability in MCF7, MDA-MB-231, 4T1-luc2, and TS/A cells were 9.35, 6.69, 24.6, and 3.74 µM, respectively.

**Figure 3 marinedrugs-21-00225-f003:**
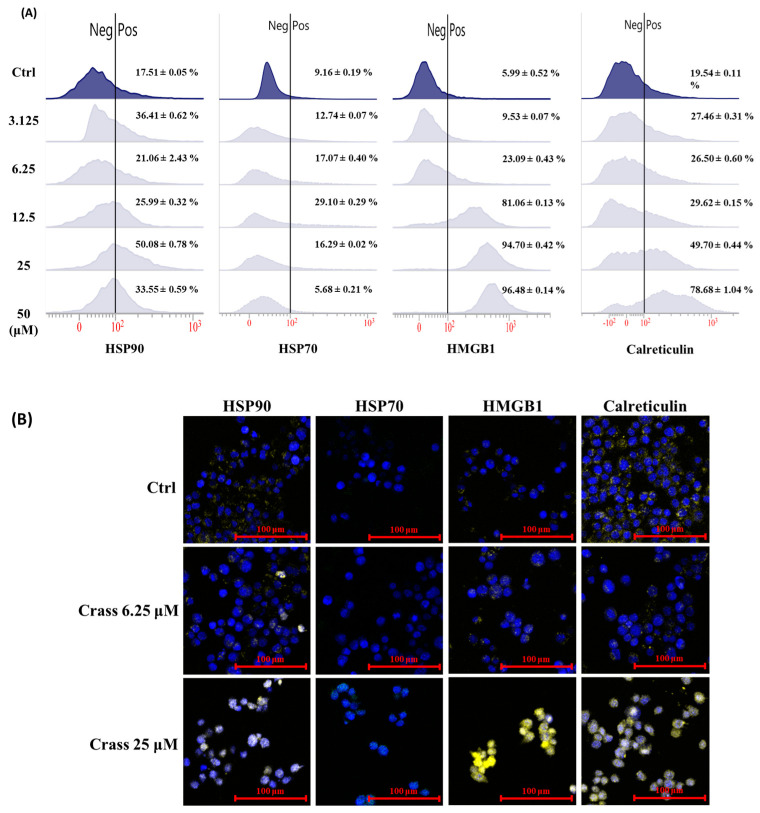
The effect of crassolide on the expression of DAMPs on the surface of 4T1-luc2 cells. The 4T1-luc2 cells were treated with increasing concentrations of crassolide for 24 h. (**A**) The expression level of DAMPs on the surface of 4T1-luc2 cells were determined by flow cytometry analysis. (**B**) Imaging of DAMP ectolocalization in crassolide-treated cells was examined under a confocal microscope. The HSP90, HSP70, HMGB1, and calreticulin proteins were stained with PE anti-HSP90, Alexa Fluor 488 anti-Hsp70, PE anti-HMGB1, and PE anti-calreticulin antibodies, respectively. The nuclei were stained with DAPI (shown in blue). The scale bar is 100 µm.

**Figure 4 marinedrugs-21-00225-f004:**
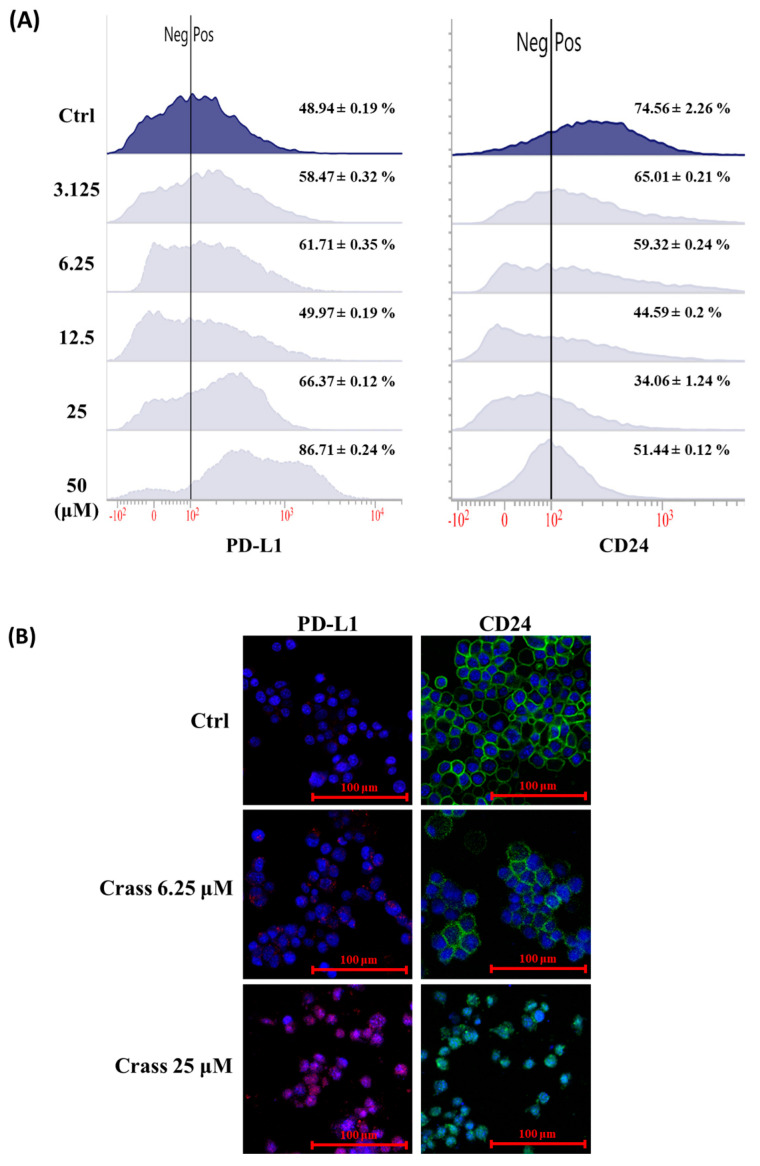
The effect of crassolide on the expression of DAMPs, PD-L1 and CD24 in 4T1-luc2 cells. The 4T1-luc2 cells were treated with increasing concentrations of crassolide for 24 h. (**A**) The expression level of PD-L1 and CD24 on the surface of 4T1-luc2 cells was determined by flow cytometry analysis. (**B**) Imaging of PD-L1 and CD24 ectolocalization in crassolide-treated cells was examined under a confocal microscope. The PD-L1 and CD24 proteins were stained with APC anti-PD-L1 and Alexa Fluor 488 anti-CD24 antibodies, respectively. The nuclei were stained with DAPI (shown in blue). The scale bar is 100 µm.

**Figure 5 marinedrugs-21-00225-f005:**
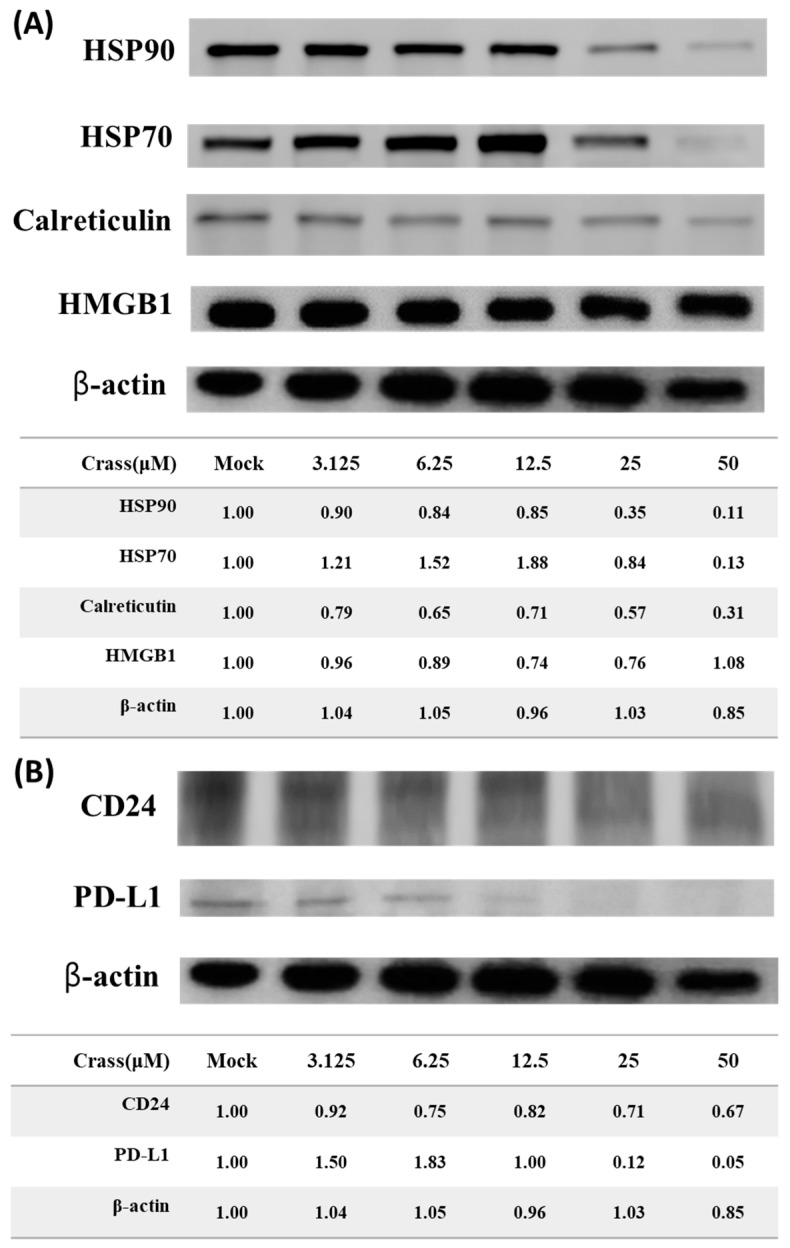
The effect of crassolide on the protein expression of DAMPs, PD-L1 and CD24 in 4T1-luc2 cells. The 4T1-luc2 cells were treated with increasing concentrations of crassolide for 24 h and the total protein was collected for assessing the expression of DAMPs, PD-L1 and CD24. (**A**) The protein expression levels of HSP90, HSP70, calreticulin and HMGB1 in total protein extracts were measured using Western blot analysis and were normalized to the protein level of β-actin. Mock: DMSO as the vehicle control. (**B**) The protein expression levels of PD-L1 and CD24 in total protein extracts were measured using Western blot analysis and were normalized to the protein level of β-actin. Mock: DMSO as the vehicle control.

**Figure 6 marinedrugs-21-00225-f006:**
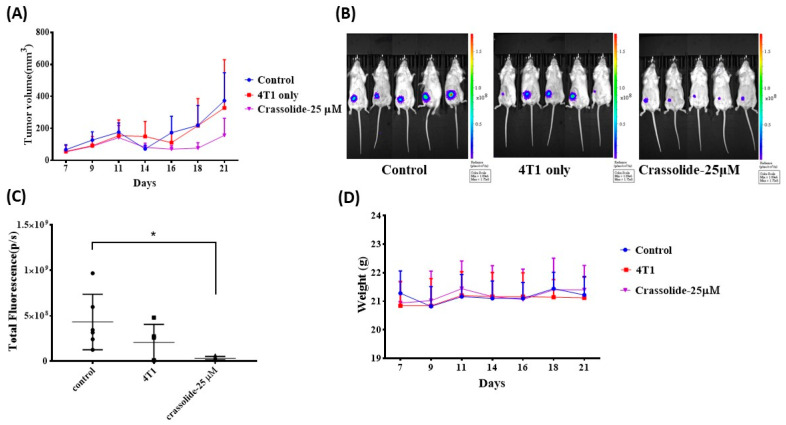
Crassolide-treated 4T1-luc2 cells effectively immunized mice against primary tumors. Test mice were vaccinated with 4T1-luc2 cells that underwent freeze-thaw cycles or were treated with crassolide at 25 µM for 24 h. At 7 d post-vaccination, live 4T1-luc2 tumor cells were implanted. Tumor progression in test mice was visualized by bioluminescence imaging (BLI). (**A**) Tumor volume was monitored until 21 d post tumor implantation. (**B**) BLI and (**C**) Quantitative data of BLI was determined at 16 d post tumor implantation. * *p* < 0.05 (**D**) Body weight of all test mice.

**Figure 7 marinedrugs-21-00225-f007:**
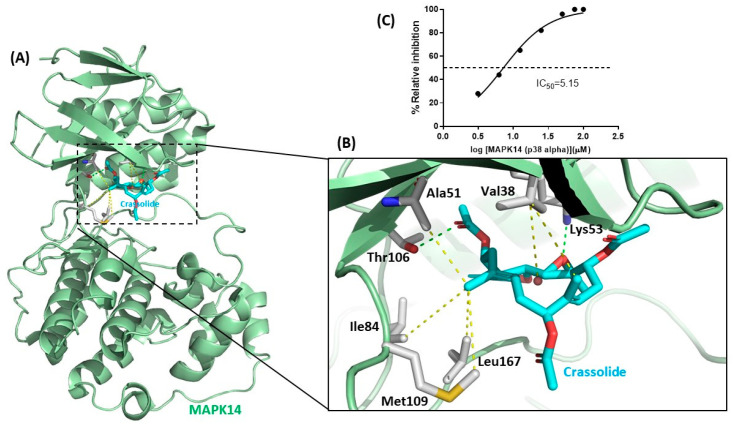
The blockade of mitogen-activated protein kinase 14 activation by crassolide. (**A**) Schematic representation of the molecular docking in the active site of MAPK14 (PDB 1DI9). The MAPK14 kinase is shown in light green cartoon. The key residues of kinase are shown in a stick representation. (**B**) The binding interactions between crassolide compound and MAPK14 are shown in hydrogen bonds (in green) and hydrophobic interactions (in yellow). (**C**) Dose–response curves of MAPK14 and its inhibitor crassolide in kinase inhibition assay.

**Figure 8 marinedrugs-21-00225-f008:**
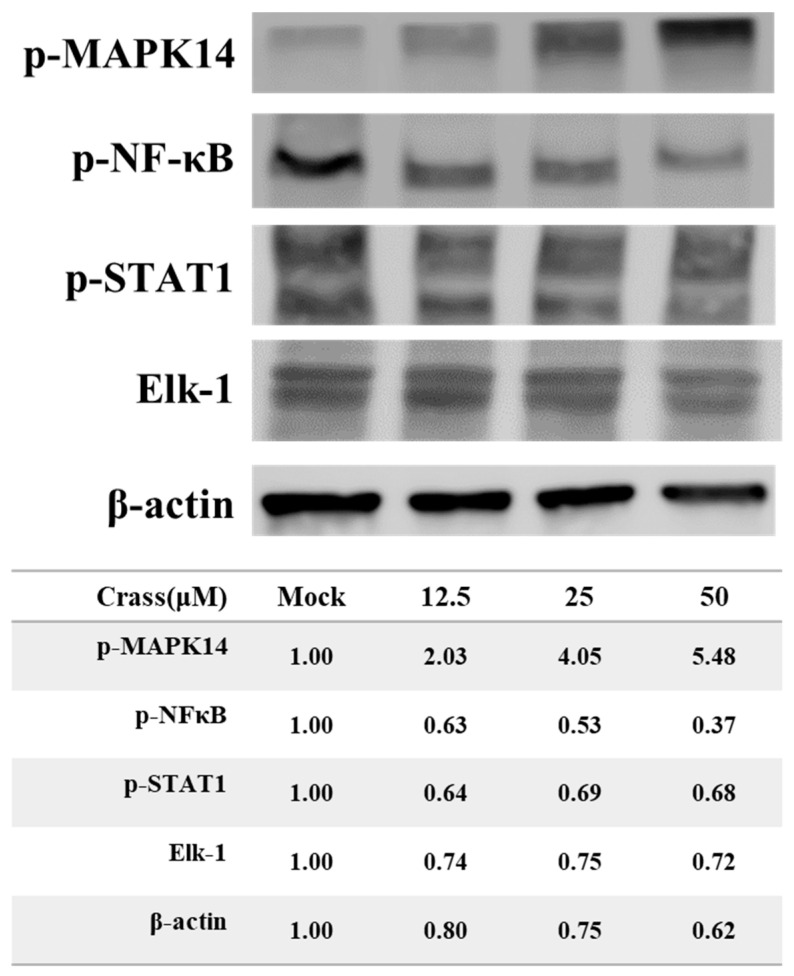
The effect of crassolide on the phosphorylation of MAPK14, NF-κB and STAT1, and EIK-1 protein in 4T1-luc2 cells. The 4T1-luc2 cells were treated with increasing concentrations of crassolide for 24 h and the total protein was collected for assessing the phosphorylation of MAPK14, NF-κB and STAT1, and EIK-1 protein using Western blot analysis and were normalized to the protein level of β-actin. Mock: DMSO as the vehicle control.

**Figure 9 marinedrugs-21-00225-f009:**
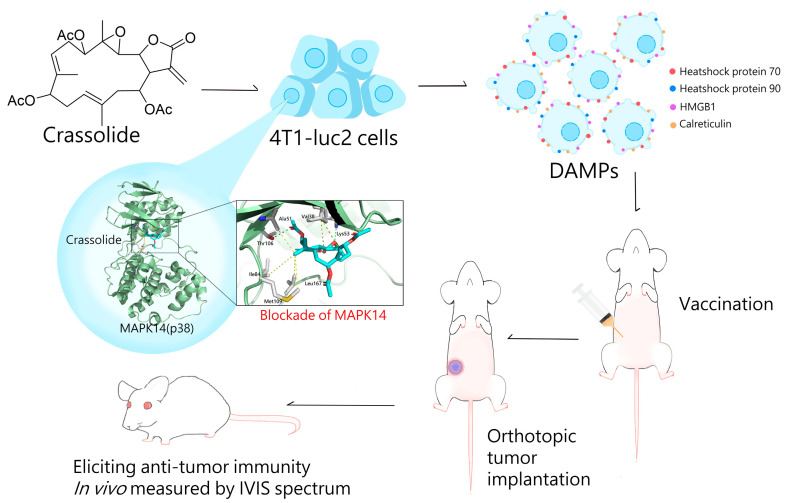
The schematic representation of crassolide triggers ICD in tumor cells and stimulates anti-tumor immunity.

**Table 1 marinedrugs-21-00225-t001:** The IC50 values of cell viability in 4T1-luc2, TS/A, MCF7, and MDA-MB-231 cells.

Cell Line	IC_50_
4T1-luc2	24.60
TS/A	3.74
MCF7	9.35
MDA-MB-231	6.69

## Data Availability

Data are available in the manuscript.

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
