# Peer review of "The Blockade of Mitogen-Activated Protein Kinase 14 Activation by Marine Natural Product Crassolide Triggers ICD in Tumor Cells and Stimulates Anti-Tumor Immunity"

_marinedrugs, 2023, doi:10.3390/md21040225_

Round 1
Reviewer 1 Report
In the manuscript “The Blockade of Mitogen-activated Protein Kinase 14 Activation by Marine Natural Product Crassolide Triggers ICD in Tumor Cells and Stimulates Anti-tumor Immunity” by Keng-Chang Tsai et al. the induction of ICD by Crassolide was demonstrated by cytotoxicity experiments, assessment of cell surface expression of specific molecules such as DAMPs, and "in vivo" antitumor activity. Interestingly, the authors were able to identify the target of crassolide, namely MAPK14, and validate its mechanism of action.
The manuscript is very interesting and the results would seem to be relevant, but some changes are needed, however.
1. In the abstract (line 31): “and other types of cancer” are mentioned, but in the manuscript only breast cancer are analyzed. This sentence needs to be modified
2. Describe the origin of Crassolide (purchased, extracted from natural sources, etc.)
3. Specify in the results, paragraph 2.1, that not all the cell lines are human and mention why transfected 4T1cells (with luc2) were used, because it does not seem useful to do the cytotoxicity on them instead of on 4T1.
4. A table with the IC50 could be useful to better define this result.
5. In fig. 3 and 4 place in the same order the protein analyzed on A and B to avoid confusion
6. In the western blotting of fig. 5 there is not present the lane showing the actin. Furthermore, in fig. 5B the same values of 5A are reported (Protein HSP90, HSP70, etc.)! This is true also for fig. 8 where are reported always the same results (Protein HSP90, HSP70, etc.). No western blotting results can be evaluated .
7. In the discussion (line 247), the sentence “In a recent study, crassolide from marine soft coral was identified as a novel inhibitor of MAPK14” has to be changed because it seems that the result was not presented for the first time in this work.
8. In M&M 4.4 specify also the other cell line used for cell viability assay
9. In M&M 4.5 what is the need to permeabilize the cells if proteins on cell surface are to be analyzed?
Author Response
Reviewer 1 Comments
Point 1: In the abstract (line 31): “and other types of cancer” are mentioned, but in the manuscript only breast cancer are analyzed. This sentence needs to be modified.
Thanks, we modified as suggested.
Point 2: Describe the origin of Crassolide (purchased, extracted from natural sources, etc.).
Thank you. We provided information on the origin of crassolide in the revised manuscript. Crassolide was obtained from Dr. Jui-Hsin Su at the National Museum of Marine Biology and Aquarium in Pingtung, Taiwan, and was isolated from the wild-type soft coral Lobophytum crissum. Please see the Materials and Methods section for further details.
Point 3: Specify in the results, paragraph 2.1, that not all the cell lines are human and mention why transfected 4T1cells (with luc2) were used, because it does not seem useful to do the cytotoxicity on them instead of on 4T1.
To investigate whether crassolide-treated tumor cells can elicit antitumor immunity by monitoring tumor luciferase activity in mice, we used 4T1-luc2 cells instead of 4T1 cells in this study. Therefore, we evaluated the effect of crassolide on cytotoxicity, DAMP production, and other assays using 4T1-luc2 cells. We apologize for incorrectly labeling 4T1 in Figure 1 and have corrected it to 4T1-luc2 cells in the revised manuscript.
Point 4: A table with the IC50 could be useful to better define this result.
As requested, we have included the IC50 values for four types of cells in Table 1 of the revised manuscript.
Point 5: In fig. 3 and 4 place in the same order the protein analyzed on A and B to avoid confusion.
As requested, we have rearranged the proteins in the same order on both panel A and B in Figures 3 and 4 of the revised manuscript.
Point 6: In the western blotting of fig. 5 there is not present the lane showing the actin. Furthermore, in fig. 5B the same values of 5A are reported (Protein HSP90, HSP70, etc.)! This is true also for fig. 8 where are reported always the same results (Protein HSP90, HSP70, etc.). No western blotting results can be evaluated.
As requested, we have presented β-actin in panel A of Figure 5 and corrected the incorrect Western blot analysis data in panel B of Figure 5 and Figure 8 of the revised manuscript.
Point 7: In the discussion (line 247), the sentence “In a recent study, crassolide from marine soft coral was identified as a novel inhibitor of MAPK14” has to be changed because it seems that the result was not presented for the first time in this work.
As per our comprehension, our study was the first to demonstrate that crassolide serves as a novel inhibitor of MAPK14. Nevertheless, if any references can provide evidence to refute our assertion, we would be glad to amend our statement.
Point 8: In M&M 4.4 specify also the other cell line used for cell viability assay.
We have provided the information about the other cell lines used for the cell viability assay in M&M 4.5 of the revised manuscript, as per your request.
Point 9: In M&M 4.5 what is the need to permeabilize the cells if proteins on cell surface are to be analyzed?
Thank you for your comment, we have corrected and provided the right protocol in M&M 4.6 of the revised manuscript.
Reviewer 2 Report
Manuscript titled " The Blockade of Mitogen-activated Protein Kinase Activation by Marine Natural Product Crassolide Triggers ICD in Tumor Cells and Stimulates Anti-tumor Immunity" is a very interesting article within the field of cancer immunotherapy research. Authors investigated the immunity-trigered anticancer mechanistic aspects for a marine-derived natural product (crassolide). The overall picture is of good quality, methods and results are easy to understand. However, there are several aspects that need to be fixed and added, including:
1. For the MTT assay neither negative nor positive controls were introduced. Additionally, positive controls are missing from all subsequent assays.
2. Safety profile of crassolide should be highlighted through testing the marine metabolite on non-cancerous human cell lines or referring reported data within this context.
3. It could be a matter of resource availability; however, authors should rationalize why they performed DAMP expressions as well as the following investigations on 4T1-luc2 cells, despite showing the modest MTT-based cytotoxic activity among other investigated cell lines.
4. Section 2.1. authors should annotate for the adopted cell lines.
5. Figures 3 and 4; the scale bars are missing from the images and figure legends.
Additionally, the description of cellular staining should be annotated at the figure legends.
6. Significance results for compared data groups (i.e. the estimated p-values) should be added within context, as well as being expressed as Asterisks on the line chat of Figure 6a.
7. Authors are advised to provide brief information regarding the MAPK14 kinase topology and canonical binding site, prior introducing the docking results.
8. Section 2.6. results introduced regarding the kinase inhibition activity, so it is proper to present IC50 values not EC50.
9. The table at Figure 8, should be texted with MAPK14, NF-κB, STAT1, and EIK-1.
10. In the discussion part, authors should thoroughly compare the cross findings, particularly the gene expression patterns, with reported with the literature for natural products of same or even close-member groups.
Author Response
Reviewer 2 Comments
1. For the MTT assay neither negative nor positive controls were introduced. Additionally, positive controls are missing from all subsequent assays.
Thanks for the reviewer’s comment. To illustrate the effect of crassolide at various concentrations on cytotoxicity and IC50 values in four different carcinoma cells, we have decided to present our data using cell survival curves with negative and positive controls. We believe that the positive controls used in all subsequent assays in our study are non-essential. However, if the reviewer insists on it, we will take the recommendation into careful consideration in the future.
2. Safety profile of crassolide should be highlighted through testing the marine metabolite on non-cancerous human cell lines or referring reported data within this context.
Thank you for the reviewer's comment. Our study highlights a novel approach by showcasing the effectiveness of the marine compound crassolide in stimulating immunogenic cell death in tumor cells in vitro and eliciting anti-tumor immunity in vivo without exposing crassolide. Therefore, in our opinion, testing crassolide on non-cancerous human cell lines to evaluate its safety is unnecessary.
3. It could be a matter of resource availability; however, authors should rationalize why they performed DAMP expressions as well as the following investigations on 4T1-luc2 cells, despite showing the modest MTT-based cytotoxic activity among other investigated cell lines.
To investigate whether crassolide-treated tumor cells can elicit antitumor immunity by monitoring tumor luciferase activity in mice, we used 4T1-luc2 cells instead of 4T1 cells in this study. Therefore, we evaluated the effect of crassolide on cytotoxicity, DAMP production, and other assays using 4T1-luc2 cells.
4. Section 2.1. authors should annotate for the adopted cell lines.
We have provided the information about the other cell lines used for the cell viability assay in M&M 4.5 of the revised manuscript, as per your request.
5. Figures 3 and 4; the scale bars are missing from the images and figure legends.
As requested, we have added a scale bar to Figures 3 and 4 of the revised manuscript.
Additionally, the description of cellular staining should be annotated at the figure legends.
As per your request, we have included a description of the cellular staining procedure in the figure legends for Figures 3 and 4 in the revised manuscript.
6. Significance results for compared data groups (i.e. the estimated p-values) should be added within context, as well as being expressed as Asterisks on the line chat of Figure 6a.
As per your request, we have added p-values within the context of the results of Section 2.5 and the figure legend for Figure 6 in the revised manuscript. However, due to the large inter-group difference in tumor volume between the control group and the experimental group, statistically significant differences could not be observed in the experimental results (Figure 6A). Therefore, we have modified the description of the results in Section 2.5 in the revised manuscript.
7. Authors are advised to provide brief information regarding the MAPK14 kinase topology and canonical binding site, prior introducing the docking results.
We have provided brief information regarding the MAPK14 kinase topology and canonical binding site in the results of Section 2.6 of the revised manuscript, as per your request.
8. Section 2.6. results introduced regarding the kinase inhibition activity, so it is proper to present IC50 values not EC50.
As per your request, we have corrected the EC50 to IC50 value in the results of Section 2.6 and Figure 7 of the revised manuscript.
9. The table at Figure 8, should be texted with MAPK14, NF-κB, STAT1, and EIK-1.
We apologize for incorrectly labeling Figure 8. We have corrected the Figure 8 labeling in the revised manuscript.
10. In the discussion part, authors should thoroughly compare the cross findings, particularly the gene expression patterns, with reported with the literature for natural products of same or even close-member groups.
As per your request, we have provided a brief discussion regarding the comparison of the cross-findings, particularly the gene expression patterns, with the literature reports for natural products of the same or even closely related groups in the discussion section of the revised manuscript.
Reviewer 3 Report
The manuscript entitled "The Blockade of Mitogen-activated Protein Kinase 14 Activa- 2 tion by Marine Natural Product Crassolide Triggers ICD in Tu- 3 mor Cells and Stimulates Anti-tumor Immunity" is well written, has overall merit and deserves to be published in "marine drugs".
The introduction is clear and provides sufficient background and include all relevant references. The discussion highlights important issues related to immunogenic cell death and the immunotherapeutic effects of crassolide in activating anticancer immune responses. The conclusions meets the results obtained.
I have only one observation referred to cell culture methodology. Authors mention only 4T1-luc2 (A) in methods section, but they workd with TS/A, MCF7 and MDA-MB-231. They must be included in 4.1 and 4.4 sections.
Author Response
Reviewer 3 Comments
The manuscript entitled "The Blockade of Mitogen-activated Protein Kinase 14 Activa- 2 tion by Marine Natural Product Crassolide Triggers ICD in Tu- 3 mor Cells and Stimulates Anti-tumor Immunity" is well written, has overall merit and deserves to be published in "marine drugs".
The introduction is clear and provides sufficient background and include all relevant references. The discussion highlights important issues related to immunogenic cell death and the immunotherapeutic effects of crassolide in activating anticancer immune responses. The conclusions meets the results obtained.
1. I have only one observation referred to cell culture methodology. Authors mention only 4T1-luc2 (A) in methods section, but they workd with TS/A, MCF7 and MDA-MB-231. They must be included in 4.1 and 4.4 sections.
Thanks for reviewer’s comments, we have provided the information about the other cell lines used for the cell viability assay in M&M 4.5 of the revised manuscript, as per your request.
Reviewer 4 Report
The manuscript, “The Blockade of Mitogen-activated Protein Kinase 14 Activation by Marine Natural Product Crassolide Triggers ICD in Tumor Cells and Stimulates Anti-tumor Immunity” by Tsai et al. investigated the effects of crassolide on the induction of ICD, the expression of immune checkpoint molecules and cell adhesion molecules, as well as tumor growth in a murine 4T1 mammary carcinoma model. This study not only highlights the immunotherapeutic effects of crassolide in activating anticancer immune responses but also provides a novel approach to developing marine natural products for clinical applications.
Comments and Suggestions for Authors:
1- Lines 2-3: It is better to mention the name of the marine species in the title.
2- Lines 16-31: In the abstract, briefly mention the materials and methods used in the study.
3- Line 32-33: Please correct the keywords according to MeSH.
4-Lines 21, 65: The scientific names of species should be italicized.
5-Lines 115, 134: Please add scale bar for figure 3.B and 4.B.
6-Line 163: In the section "4.3. Animal model" it was said that the studied mice were divided into 4 groups. In the results section of "2.5. Crassolide-treated 4T1-luc2 cells effectively immunized mice against primary tumors" there is no mention of the fourth group (treated with crassolide 6.25 μM). Please explain this.
7- Line 216: In the discussion section, please mention the limitations of this study in a separate paragraph.
8- Lines 288, 277: The number of mice studied should be mentioned. It should also be determined how many mice are in each group.
9-Lines 310-311: Please indicate the source and how the crassolide was prepared.
Author Response
Reviewer 4 Comments
The manuscript, “The Blockade of Mitogen-activated Protein Kinase 14 Activation by Marine Natural Product Crassolide Triggers ICD in Tumor Cells and Stimulates Anti-tumor Immunity” by Tsai et al. investigated the effects of crassolide on the induction of ICD, the expression of immune checkpoint molecules and cell adhesion molecules, as well as tumor growth in a murine 4T1 mammary carcinoma model. This study not only highlights the immunotherapeutic effects of crassolide in activating anticancer immune responses but also provides a novel approach to developing marine natural products for clinical applications.
Comments and Suggestions for Authors:
1. Lines 2-3: It is better to mention the name of the marine species in the title.
Thank you for the reviewer's suggestion. However, including the name of the marine species in the title may make it lengthy. Therefore, we have decided to retain the current title. If the reviewer persists in recommending such a change, we will take it into careful consideration and make modifications as necessary.
2. Lines 16-31: In the abstract, briefly mention the materials and methods used in the study.
As requested, we have modified the abstract of the revised manuscript to briefly mention the materials and methods used in the study.
3. Line 32-33: Please correct the keywords according to MeSH.
We have corrected the keywords in the revised manuscript as per the request, aligning them with MeSH.
4. Lines 21, 65: The scientific names of species should be italicized.
Thank you for bringing this to our attention. We apologize for the oversight and have made the necessary changes to italicize the scientific names of species throughout the manuscript.
5. Lines 115, 134: Please add scale bar for figure 3.B and 4.B.
As requested, we have added a scale bar to Figures 3 and 4 of the revised manuscript.
6. Line 163: In the section "4.3. Animal model" it was said that the studied mice were divided into 4 groups. In the results section of "2.5. Crassolide-treated 4T1-luc2 cells effectively immunized mice against primary tumors" there is no mention of the fourth group (treated with crassolide 6.25 μM). Please explain this.
We apologize for providing wrong information in section 4.3 ("Animal model"), where we stated that the mice were randomly assigned to four groups, and group 4 was injected with 2x106 of 4T1-luc2 cells that had been treated with crassolide (6.25 µM) for 24 hours. In fact, we did not test group 4 (treated with crassolide 6.25 μM). We have corrected this mistake in the revised manuscript.
7. Line 216: In the discussion section, please mention the limitations of this study in a separate paragraph.
As per your request, we have included our perspective on the limitations of this study in the discussion section of the revised manuscript.
8. Lines 288, 277: The number of mice studied should be mentioned. It should also be determined how many mice are in each group.
As per your request, we have included in the revised manuscript the total number of mice studied and their respective sizes in each group.
9. Lines 310-311: Please indicate the source and how the crassolide was prepared.
Thank you. We provided information on the origin of crassolide in the revised manuscript. Crassolide was obtained from Dr. Jui-Hsin Su at the National Museum of Marine Biology and Aquarium in Pingtung, Taiwan, and was isolated from the wild-type soft coral Lobophytum crissum. Please see the Materials and Methods section for further details.
Round 2
Reviewer 1 Report
In this version the manuscript is ready for the publication.
Just a comment for the authors on the point 7 of my report: I strongly believe that your results are new. I only suggest that this sentence be changed because in this version it seems to refer to a "recent study" by other authors. You could say "in this study".
Author Response
Just a comment for the authors on the point 7 of my report: I strongly believe that your results are new. I only suggest that this sentence be changed because in this version it seems to refer to a "recent study" by other authors. You could say "in this study".
As requested, we have changed "In a recent study" to "In this study" in the discussion of the revised manuscript.
Reviewer 2 Report
The authors kindly responded to almost all suggestions and comments. The manuscript has been comprehensively presented within its current form. However, in my opinion, the authors should still highlight the safety profile of crassolide even through citing reported data within current literature.
Author Response
The authors kindly responded to almost all suggestions and comments. The manuscript has been comprehensively presented within its current form. However, in my opinion, the authors should still highlight the safety profile of crassolide even through citing reported data within current literature.
Thank you for your kind words and useful suggestion. As requested, we have added information about the safety profile of crassolide to the discussion section of the revised manuscript."
Reviewer 4 Report
Thank you for considering my comments and for corrections.
Author Response
Thank you for considering my comments and for corrections.
Again, thank you very much for your useful suggestion.